# Tumor Progression and Treatment-Related Changes: Radiological Diagnosis Challenges for the Evaluation of Post Treated Glioma

**DOI:** 10.3390/cancers14153771

**Published:** 2022-08-03

**Authors:** Danlei Qin, Guoqiang Yang, Hui Jing, Yan Tan, Bin Zhao, Hui Zhang

**Affiliations:** 1College of Medical Imaging, Shanxi Medical University, Taiyuan 030001, China; qindl1204@163.com; 2Shanxi Province Key Laboratory of Oral Diseases Prevention and New Materials, Shanxi Medical University School, Hospital of Stomatology, Taiyuan 030001, China; 3Department of Radiology, First Clinical Medical College, Shanxi Medical University, Taiyuan 030001, China; doctor_ygq@163.com (G.Y.); tanyan123456@sina.com (Y.T.); 4Department of MRI, The Six Hospital, Shanxi Medical University, Taiyuan 030008, China; doctor511099@163.com; 5Intelligent Imaging Big Data and Functional Nano-imaging Engineering Research Center of Shanxi Province, Taiyuan 030001, China

**Keywords:** glioma, treatment-related changes, tumor progression, magnetic resonance imaging, positron emission tomography, artificial intelligence

## Abstract

**Simple Summary:**

Glioma is the most common primary malignant tumor of the adult central nervous system. Despite aggressive multimodal treatment, its prognosis remains poor. During follow-up, it remains challenging to distinguish treatment-related changes from tumor progression in treated patients with gliomas due to both share clinical symptoms and morphological imaging characteristics (with new and/or increasing enhancing mass lesions). The early effective identification of tumor progression and treatment-related changes is of great significance for the prognosis and treatment of gliomas. We believe that advanced neuroimaging techniques can provide additional information for distinguishing both at an early stage. In this article, we focus on the research of magnetic resonance imaging technology and artificial intelligence in tumor progression and treatment-related changes. Finally, it provides new ideas and insights for clinical diagnosis.

**Abstract:**

As the most common neuro-epithelial tumors of the central nervous system in adults, gliomas are highly malignant and easy to recurrence, with a dismal prognosis. Imaging studies are indispensable for tracking tumor progression (TP) or treatment-related changes (TRCs). During follow-up, distinguishing TRCs from TP in treated patients with gliomas remains challenging as both share similar clinical symptoms and morphological imaging characteristics (with new and/or increasing enhancing mass lesions) and fulfill criteria for progression. Thus, the early identification of TP and TRCs is of great significance for determining the prognosis and treatment. Histopathological biopsy is currently the gold standard for TP and TRC diagnosis. However, the invasive nature of this technique limits its clinical application. Advanced imaging methods (e.g., diffusion magnetic resonance imaging (MRI), perfusion MRI, magnetic resonance spectroscopy (MRS), positron emission tomography (PET), amide proton transfer (APT) and artificial intelligence (AI)) provide a non-invasive and feasible technical means for identifying of TP and TRCs at an early stage, which have recently become research hotspots. This paper reviews the current research on using the abovementioned advanced imaging methods to identify TP and TRCs of gliomas. First, the review focuses on the pathological changes of the two entities to establish a theoretical basis for imaging identification. Then, it elaborates on the application of different imaging techniques and AI in identifying the two entities. Finally, the current challenges and future prospects of these techniques and methods are discussed.

## 1. Introduction

Gliomas are the most common primary malignant brain tumors in adults and are associated with a dismal prognosis. The median overall survival (OS) for glioblastoma (GBM) ranges from 14 to 18 months [1,2]. In clinics, the standard of care for high-grade glioma (HGG) consists of surgical resection, followed by concurrent radiotherapy and adjuvant temozolomide (TMZ) chemotherapy. For low-grade glioma (LGG), molecular subtyping plays an important role in patient prognosis and selection of treatment, including surgical resection, radiotherapy, or radiotherapy followed by PCV (procarbazine, lomustine and vincristine) or temozolomide chemoradiotherapy. Moreover, the use cycle of TMZ and molecular typing jointly affect the prognosis and survival of patients [3]. During follow-up, the appearance of new and/or increasing enhancing lesions on conventional MRI after the completion of chemoradiation therapy has become a major challenge because it may represent tumor progression (TP) or treatment-related changes (TRCs). TRCs are histopathologically characterized by fibrinoid necrosis of blood vessel walls, blood–brain barrier(BBB)disruption, oligodendroglial injury and cellular hypoxia, leading to increased vessel permeability and edema [4,5]. The pathological features of TP include neovascularization, tumor cell proliferation and neoplastic BBB disruption. Because of the difference in pathological basis, the clinical management strategies of these two entities may be completely different. Patients with TP may require reoperation or a change of their treatment plan, while those diagnosed with TRCs need to be closely monitored through short-interval MRI scans to support the continuation of currently effective therapy for better clinical outcomes. An incorrect diagnosis of TP may lead to the erroneous termination of effective treatment and has a negative impact on survival. In addition to affecting individual patient care, these changes also have an impact on the results of clinical trials of new therapies. Thus, precisely detecting TRCs and TP is critical for ensuring effectiveness of a treatment. Although histopathological biopsy is the “gold standard” for diagnosing TP and TRCs, its accuracy depends on the biopsy site, resection type, and lesion heterogeneity. Additionally, its invasiveness limits its clinical application. To overcome these challenges, non-invasive advanced imaging methods have been developed, which has made distinguishing TRCs from TP at the early stage possible. Thus, these methods play an essential role in clinical decision-making regarding treatment choice and management of patients.

As we all know, the phenomenon of conventional MRI enhancement highlights that enhancement does not measure the tumor activity, but only reflects the disturbed BBB. Considering the different pathological mechanisms of the two entities, numerous innovative imaging modalities focusing on cell proliferation (apparent diffusion coefficient parameters from diffusion techniques), neoangiogenesis and cerebral blood flow (from perfusion techniques) or metabolism (from MRS, APT, or PET) are valuable diagnostic tools. These methods are valuable in both the baseline and follow-up evaluations of brain tumors., As an emerging field, MRI-based AI can reflect tumors heterogeneity and occupies crucial position in the evaluation of the tumor microenvironment after treatment. These methods allow considerably deeper and non-invasive insight into the interpretation of brain lesions, thereby improving the accuracy of diagnosis.

Here, we discuss the potential value of imaging techniques in distinguishing between TP and TRCs of gliomas. First, we outline the pathological features of the two entities to provide a theoretical basis for the subsequent imaging identification. Then, we focus on the main application of different imaging techniques and MRI-based AI in tracking TP and TRCs. Furthermore, we present the post-processing diagrams to distinguish between the two entities using different technologies and workflow (Figure 1). Non-invasive in vivo imaging may be promising in clinical decision-making. Finally, we propose the current challenges and future prospect of these techniques and AI.

## 2. Characteristics of TRCs

### 2.1. Pseudoprogression (PsP)

Based on the timing and degree of injury, TRCs may be clinically categorized into early pseudoprogression (PsP) and late radiation necrosis (RN) [6]. PsP is predominantly a subacute treatment-related reaction that may or may not involve neurological deterioration [4]. Radiologically, PsP can be described as new or enlarging areas of contrast enhancement on follow-up MRI within the first 3–6 months of the completion of chemoradiotherapy following surgical resection, which subsides or stabilizes without further intervention treatments, and some studies suggest that PsP may have a relatively good prognosis [7,8]. Various reports have defined the time of PsP occurrence differently, which is likely to affect the incidence. Previous studies have shown that the incidence rate of PsP after chemoradiotherapy ranges from 21 to 47% in GBM patients treated with TMZ [4,8,9,10]. The O^6^-methylguanine-DNA methyltransferase (MGMT) status [9], isocitrate dehydrogenase (IDH) gene status [11], Ki67 expression [12], and p53 status [13] were all associated with PsP occurrence. Tumors with a methylated MGMT promoter and IDH mutation show PsP more frequently and have better median OS [7,9], and a study found a 91.3% probability of PsP in patients with methylated MGMT promoter tumors [9]. These biomarkers clearly affect PsP incidence and prognosis. The PsP pathophysiology remains unclear. At the histological and mechanistic level, early PsP and RN may represent different pathophysiological processes, where some patients with early PsP continue to develop trueRN, whereas others may show improvement [1]. PsP may constitute an over-response to effective therapy and is likely associated with endothelial damage, BBB disruption, oligodendroglial injury, cellular hypoxia and the upregulation of the vascular endothelial growth factor (VEGF), leading to increased vessel permeability and edema or increased tumor enhancement; another explanation is the increased capillary permeability induced by radiotherapy [4]. The exact pathophysiological features of PsP and the associated molecular changes need further exploration.

### 2.2. Radiation Necrosis (RN)

In patients with malignant gliomas, RN is a severe local tissue reaction to radiotherapy. Classical RN represents a variant of post-treatment effects, which is different from PsP in terms of the time of onset and degree of severity. RN is usually an undesirable but inevitable effect of radiotherapy. It generally occurs 6–24 months after radiotherapy but can occur up to several years or even decades after [14,15]. RN commonly presents as a space occupying necrotic mass provoking neurological deficit. Because of differences in the radiation dose and fractionation, target lesion volume, and the time of reporting RNfrom radiotherapy, the incidence of RN ranges from 5 to 40% [16,17]. The pathological features of RN include endothelial cell damage, vascular dilation and telangiectasias, wall thickening and vessels hyalinization, fibrinoid necrosis of blood vessel walls, and adjacent perivascular parenchymal coagulative necrosis [4,5]. Radiation therapy also damages astrocytes, oligodendrocytes, and oligodendrocyte progenitor cells [17]. Vasogenic edema and hypoxia cause the up-regulation and release of hypoxia-inducible factor-1a (HIF-1a), tumor necrosis factor-alpha (TNF-α) and VEGF, which can induce apoptosis of endothelial cells and increase the permeability of small vessels and BBB [18]. A complete understanding of the pathophysiology and molecular pathways of RN is useful for comprehending and interpreting conventional and advanced imaging findings.

### 2.3. Pseudoresponse

Anti-VEGF agents (such as bevacizumab) have recently been used for HGG treatment trials, which produce “normalization” of the BBB. This is a direct effect on blood vessel permeability, rather than a true anti-tumor reduction [2]. Anti-VEGF agents result in a marked pattern of change on MRI. A rapid decrease in contrast enhancement and a decrease in the surrounding edema are observed on fluid-attenuated inversion recovery (FLAIR) [5]. This radiological response should be interpreted with caution, as it lasts just a few days or weeks, and therefore is termed “pseudoresponse”. In clinics, this response can relieve symptoms, reduce steroid dependence and improve the quality of life for patients [14]. However, it was pointed out that this normalization of blood vessels is reversible, with rebound enhancement and edema, and may affect the prognosis of patients.

TP is characterized by the presence of tumor cells, increased cellularity, and vascular proliferation, leading to BBB disruption, whereas the pathological features of TRCs include vascular endothelial cell damage, BBB disruption, and cellular hypoxia, leading to edema and increased vascular permeability. Because of the difference in pathological changes between TP and TRCs, it provides a theoretical basis for imaging identification. Thus, we describe the diagnostic value of different imaging modalities for predicting the TP and TRCs.

## 3. Diagnostic Imaging Modalities

### 3.1. Conventional MRI

At present, MRI plain scan and enhanced scan are the most crucial imaging indicators for evaluating the efficacy and recurrence of gliomas. In 1990, the Macdonald criteria [19] were the most widely used guidelines for assessing response to therapy in patients with HGG. TP is considered to have occurred only when the perpendicular diameters of the largest area of contrast enhancement lesion increased to >25%, any new tumor appeared on computed tomography (CT) or MRI, or clinical deterioration occurred. The development of MRI technology and the observation of number of clinical cases have clarified that the size of MRI contrast enhancement lesions can no longer accurately determine the tumor recurrence basis. Because the size of the enhanced lesions only reflect the extent of permeability enhancement after BBB destruction, but do not represent the of tumor volume size. Increased enhancement can be induced by various non-tumoral processes, such as treatment-related inflammation, ischemia, subacute radiation effects, and RN [14]. Subsequently, in 2010, a more comprehensive Response Assessment in Neuro-Oncology (RANO) guideline for evaluating the glioma treatment efficacy was proposed [20]. The disease progression was determined according to MRI characteristics and the time after chemotherapy (less than 12 weeks and >12 weeks). The enhancement outside the radiation field, the increase of 25% or more of the sum of the vertical diameter product between the first radiotherapy scan and the scan after 12 weeks, or clinical deterioration are considered to be important factors for TP. For patients receiving antiangiogenic therapy, a significant increase in T2-FLAIR non-enhancing lesion may be considered to indicate progressive disease [20]. The modified RANO criteria were published in 2017; according to these criteria, TP was defined based on at least two sequential scans separated by ≥4 weeks, both exhibiting a ≥ 25% increase in the sum of products of perpendicular diameters, a ≥ 40% increase in the total volume of the enhancing lesions or clear clinical deterioration. Given the existence of PsP, a repeat study needs to be conducted 4 weeks later to confirm any new measurable (>10 mm × 10 mm) enhancing lesion r [21]. In fact, within the first 12 weeks after treatment, the new lesions with enhancement seen within the radiation field can never be definitively diagnosed as progression versus PsP.

Studies have shown that the enhancement mode, enhancement location, and enhancement size of lesions can help to identify TP and TRCs. For instance, the typical appearance of RN on contrast-enhanced T1 MRI is the so-called Swiss cheese or soap bubble [22]. Another study demonstrated that 85% of patients with TP confirmed by histopathology observed focal solid nodular enhancement and solid uniform enhancement with distinct margins, while patients with RN could see a hazy mesh-like diffuse enhancement and rim enhancement with feathery indistinct margins [23]. In some retrospective studies, distant subependymal enhancement [24,25], corpus callosum combination with multiple enhancing lesions and crossing of the midline [26] can provide clues for early TP diagnosis. A study showed that larger sizes of T2-FLAIR signal abnormality and the enhancing component of the lesion may favor TP, likely reflecting greater disruption of the BBB [27]. Although enhancing T1 and FLAIR can show subtle changes in the lesion, its sensitivity and specificity for TP detection are relatively lower.

Differentiating TP from TRCs based on a two-dimensional (2D) measurement of the enhancing area and the limited information obtained through conventional MRI is currently challenging. Compared with conventional MRI, advanced imaging techniques provide a plethora of additional parameters, which might result in higher levels of diagnostic performance for TP and TRCs. The integration of additional information from advanced neuroimaging techniques may further improve the diagnostic accuracy of conventional MRI. Next, we discuss the application value of advanced imaging techniques in distinguishing the two entities and elaborate on the advantages and disadvantages of each technique. Table 1 provides a summary of current techniques and their advantages and limitations.

### 3.2. Diffusion MRI

#### 3.2.1. Diffusion-Weighted Imaging

Diffusion-weighted imaging (DWI), reflecting the microscopic movement of water molecules in tissues, has been widely used for evaluating tumor grading and treatment response or disease progression. Apparent diffusion coefficient (ADC), as a very accessible parameter of DWI, reflects microscopic water diffusivity in the presence of factors that limit diffusion in tissues [37]. In HGGs, as cellularity is increased, the tissue is detected as restriction diffusion and the ADC value decreases. Several studies have shown that the ADC value is s useful for differentiating TRCs from TP in treated HGGs [38,39,40,41,42]. Through qualitative analysis, Lee et al. [41] found that the occurrence rate of homogeneous or multi-focal high signal intensity of TP on DWI is higher than that of PSP, and a mean ADC value lower than 1200 × 10^−6^ mm^2^/s was more common in TP than in PsP. In addition to mean ADC value quantitative and qualitative analyses, cumulative histogram analysis [25,43,44] and parametric response maps [45] were novel approaches for differentiating TP and PsP. Although diffusion MRI with an ADC value is the most commonly used advanced technology at present, several meta-analyses have demonstrated that diffusion MRI is not suitable for differentiating TP from RN when used alone, and its diagnostic accuracy is the lowest among all advanced MRI techniques [46,47]. Due to the spatial and genetic heterogeneity of HGG and the complex pathology of the surgical area after radiotherapy, it will have a large impact on ADC values, leading to large deviations in research results [16,48]. Moreover, ADC values may be affected by the IDH status, with the ADC value of IDH wild-type gliomas being lower than that of IDH mutant gliomas [48]. Another limitation is that perfusion can interfere with accurate diffusion measurements due to incoherent movement of blood.

#### 3.2.2. Intravoxel Incoherent Motion

To overcome the limitations of DWI, intravoxel incoherent motion (IVIM) based on DWI is proposed. The IVIM technology can obtain tissue diffusion and perfusion characteristics simultaneously and uses a biexponential model for diffusion calculation [49]. AM Paschoal et al. [50] opined that IVIM could help in distinguishing between the diffusion of water molecules and micro-circulation perfusion in tissues. The IVIM model involves two diffusion coefficients. The slow diffusion coefficient (D) is related to molecular diffusion restriction, whereas the fast diffusion coefficient (D*) is related to blood movement in the micro-vascular system. The perfusion fraction (f), the third parameter, describes the fraction of incoherent signals generated from the vascular compartment in each voxel. Several studies have used IVIM to evaluate the grade [51], prognosis [52,53] and treatment response [49] of gliomas. Kim et al. investigated 51 patients with pathologically confirmed glioma progression (31 cases) and TRCs (20 cases) by using IVIM-DWI scans and found that the mean 90th percentile for perfusion was significantly higher in the recurrent tumor group than in the TRC group [49]. A prospective study investigated the combination of IVIM and arterial spin labeling (ASL) for differentiating TP from PsP [54]. IVIM perfusion imaging has several theoretical advantages. It can provide diffusion and perfusion information simultaneously without the need of intravenous contrast agents. IVIM is in valuable for some patients with renal failure or allergic reactions. At the same time, IVIM can be easily scanned repeatedly because the number of b-values collected is small and the total collection time is short, which is well-accepted by both patients and clinicians. Most studies have used 13–16 b-values to acquire IVIM images, and the average scan time has been reported to vary from 3 to 5 min [49,52,53,54]. Various factors may affect the quality of IVIM brain perfusion imaging, including the selection of b value [49], low signal to noise ratio at the high b value, partial-volume contamination from the cerebrospinal fluid, large-vessel partial, necrotic areas, and susceptibility artifacts [53]. Thus, the exact relationship between IVIM perfusion parameters and dynamic susceptibility contrast (DSC) perfusion parameters should be investigated further.

#### 3.2.3. Diffusion Tensor Imaging

Based on the Gaussian diffusion model, diffusion tensor imaging (DTI) can measure the directional variation of water diffusivity and predict the structural properties of tissues [55]. DTI is often used to track the integrity of white matter fiber bundles and assess the degree of white matter damage and recovery after chemoradiotherapy. Fractional anisotropy (FA), relative anisotropy (RA) and mean diffusivity (MD) are the commonly used parameters in DTI [55]. Because brain white matter fiber bundles are extensively damaged, with almost no normal fibers and cell structures, after radiation injury, the FA value of PsP or RN is considerably much lower than that of TP. A study reported that higher MD values and lower FA metrics in both enhanced lesion and related edema of TRCs were significantly different compared with those of TP [56]. Several studies have also pointed out that the FA ratio in RN or PsP was lower than that in TP [57,58]. However, another retrospective study showed that DTI metrics were unavailable in differentiating TP and PsP compared with morphologic MRI features. They thought that DTI indicators were affected by several factors, such as extracellular volume, tumor cell density and direction [27]. Technically, the result of DTI also depends on the choice of b value and direction number. To achieve a more accurate diffusion tensor, the number of DTI acquisition directions is usually increased. The common number of directions ranges from 15 to 64, and the acquisition time is 5~30~min; this will increase the impact of patient motion artifacts [55]. Additionally, because of the long collection time, the feasibility of using DTI in clinical practice is poor. Thus, compared with DWI and IVIM, DTI is less commonly used in clinical research to evaluate treatment response.

#### 3.2.4. Diffusion Kurtosis Imaging

In a complex biological tissue microenvironment, the actual diffusion of water molecules shows different degrees of non-Gaussianity. Diffusion kurtosis imaging (DKI) is a straightforward extension of DWI that captures the non-Gaussian water molecule diffusion behavior as a reflective marker for tissue heterogeneity [59]. DKI parameters have recently been used as potential imaging biomarkers to grade gliomas [60], predict its genotype [61] and distinguish HGG and solitary brain metastasis [62]. Recent studies have shown that DKI may be a reliable tool in differentiating TP from TRCs. Research shows that relative MK (rMK) was significantly higher in the TP group than in the PsP group (the AUC was 0.914 for rMK, with an 85% diagnostic accuracy), and rMK appeared to be the best independent predictor [63]. In addition, DKI combined with DSC MRI can improve diagnostic performance in assessing treatment response compared with either technique alone and the diagnostic accuracy to 88.24% [64]. However, DKI is a multi-b value multi-directional collection, and the collection time is longer than DTI. No standard process exists for the selection of b value and direction setting of DKI.

In general, the clinical application of each diffusion technology is not exactly the same. Several factors will affect the measurement of DWI-ADC value. As a single imaging measurement, DWI seems to have sufficient sensitivity but perhaps inadequate specificity. IVIM can provide information of tumor tissue diffusion and perfusion at the same time. DTI has high sensitivity and specificity in showing the structural integrity of white matter. It is a method for detecting invasive tumors. DKI can well characterize and reflect the complexity and heterogeneity of tumor tissue microenvironment, and MK is the most meaningful parameter. Thus, we noticed that compared with DWI and DTI, IVIM and DKI technology may have more potential added value for identifying TP and TRCs and can provide useful structural information of microcirculation perfusion and tumor microenvironment. Although promising, we believe that the diagnostic performance of DKI and IVIM needs further research before DKI and IVIM technology is incorporated into the routine clinical workflow of neuro-tumor application.

### 3.3. Perfusion MRI

Glioma is a highly vascular tumor characterized by endovascular proliferation and angiogenesis. Perfusion MRI is one of the most widely used imaging techniques to evaluate treatment response and has higher diagnostic accuracy. Due to their different pathological basis, TP and TRCs exhibit different perfusion characteristics. The most frequently used perfusion MRI techniques include DSC imaging, dynamic contrast-enhanced (DCE) imaging and ASL.

#### 3.3.1. Dynamic Susceptibility Contrast (DSC)

DSC is a T2*-weighted magnetic sensitive dynamic contrast enhanced MR perfusion imaging with fast acquisition speed and simple post-processing compared with other perfusion techniques. DSC utilizes the T2* effect of the paramagnetic contrast agent that causes a transient decrease in signal intensity during the initial pass through the vasculature by creating a local magnetic field distortion around the vessels. Commonly used indicators are relative cerebral blood volume (rCBV) and relative cerebral blood flow (rCBF) and mean transit time (MTT) [65]. Two meta-analyses on perfusion MRI for evaluating glioma treatment response revealed that DSC is the most commonly applied perfusion method in clinical practice [66,67].

Many DSC studies have shown that mean rCBV [68] and maximum rCBV [58] were lower in areas of RN or PsP than in those of TP. In addition to the aforementioned parameters, rCBV histogram analysis [69] and the percentage change of skewness and kurtosis of normalized rCBV [70] have been proposed to be useful for differentiating TP from TRCs. Each study generates a different threshold to identify the two entities. A recent meta-analysis showed that the threshold range of CBV was wide, with a mean rCBV ratio of 0.90–2.15 and a maximum rCBV ratio of 1.49–3.10 [66]. Due to the complexity of pathological tissue components and the lack of standardization of post-processing of MR perfusion imaging, differences are observed in the reported cut-off values. Subsequently, many studies have used the change trend of rCBV before and after treatment for differential diagnosis. A study analyzing the baseline rCBV and the rCBV maps of gliomas with concurrent chemoradiotherapy reported that changes in rCBV over time were predictive; that is, PsP had an overall negative linear trend in rCBV and TP had a positive slope [71]. DSC-MRI has been widely applied for the preoperative classification of gliomas and the differential diagnosis of intracranial tumors because of its fast acquisition speed and simple post-processing. However, DSC has a poorer spatial resolution and is easily affected by susceptibility artifacts of large blood vessels and bones [72,73]. The local inflammatory response of lymphocyte and macrophage infiltration may increase rCBV [74]. The DSC analysis assumes that the contrast material remains in the blood vessel (i.e., the BBB is intact). In fact, the tumor-induced destruction of the BBB causes the leakage of the contrast agent, thereby making the DSC quantitative results inaccurate. Thus, the leakage of contrast media must be considered.

#### 3.3.2. Dynamic Contrast-Enhanced

DCE T1-weighted perfusion MRI is capable of obtaining quantitative pharmacokinetic parameters of the tumor microcirculation structure and function. Tissue blood perfusion and neovascular permeability can be more accurately assessed by fitting a two-compartment hemodynamic model. DCE images can calculate quantitative parameters, including K^trans^ (the volume transfer coefficient from the plasma to extracellular space), V_e_ (also known as the leakage space) and V_P_ (the plasma space volume per unit tissue volume) [75]. The semi-quantitative parameter is the area under the initial curve (IAUC), which represents the amount of change in signal intensity with time when the contrast agent enters and stays in blood vessels and tissues and reflects the tissue blood volume.

Model-based pharmacokinetic DCE parameters can be used to measure real physiological mechanisms, such as blood flow and endothelial permeability [76]. Several retrospective [77] and prospective [78] studies have confirmed that K^trans^ and V_p_ were significantly different between TP and PsP. For example, Yun et al. and Thomas et al. [77,79] performed DCE-MRI in patients with GBM and found that the mean K^trans^, V_p_ and V_e_ were higher in TP than in PsP. However, the results of another study are inconsistent with those of the previous research. They confirmed that patients with PsP had significantly higher K^trans^ values than patients with TP [75]. Different pharmacokinetic models and study sample sizes lead to different of K^trans^ measurement results. The pharmacokinetic model is affected by many factors, including parameter coupling, arterial input function, water exchange, and model fitting instability. Model-free semi-quantitative parameters (IACU) can overcome the aforementioned limitations and evaluate the treatment response. One study suggested that the bimodal histogram parameters of the mean area under the time signal–intensity curves ratio at a higher curve (mAUCR_H_) is the best predictor of PsP, with higher sensitivity and specificity [76]. DCE-MRI is considerably less susceptible to artifacts, and its high spatial resolution allows accurate characterization of the vascular microenvironment of lesions [73,77]. Longer scanning time, reduced temporal resolution, complex post-processing and quantification of images are the disadvantages of DCE. Because of the lack of uniformity in data acquisition and the complexity of pharmacokinetic models, DCE imaging cannot be widely used in clinics [76]. This technology needs to be further optimized.

#### 3.3.3. Arterial Spin Labeling (ASL)

ASL is a noninvasive perfusion MRI technique wherein inflowing blood is labeled magnetically and does not dependent on exogenous contrast agents. At present, the main implementation methods of ASL are pulse labeling and pseudo-continuous labeling [80]. The CBF is the most frequently generated parameter. Studies focusing on PsP or RN specifically using ASL are scarce. In a 21-patient retrospective study, Ye et al. [81] confirmed that the normalized ASL-CBF ratio and DSC-rCBV ratio were significantly higher in TP than in radiation injury. ASL-derived CBF values were well-correlated with rCBV values obtained through DSC perfusion imaging. Choi et al. [82] showed that ASL and DSC perfusion imaging in combination are advantageous in diagnosing PsP and early TP compared with DSC alone. Another study suggested that ASL is superior to DSC and fluorine-18-fluorodeoxyglucose positron emission tomography (^18^F-FDG PET) in distinguishing TP from RN or PsP [83]. Three-dimensional pseudocontinuous ASL (3D-pCASL), as a newly developed ASL sequence, combines the advantages of pulsed and continuous labeling and 3D data acquisition, which have been used to distinguish TRCs from TP in glioma patients [54,84]. Contrast-free 3D-pCASL is a suitable alternative to DSC-MRI and can be used for long-term follow-up of postoperative radiotherapy patients with gliomas. Compared with other perfusion techniques, a major advantage of ASL is the avoidance of leakage effects with BBB disruption, which allows for more accurate quantification of CBF [80]. ASL is a good choice for patients with kidney damage and contrast agent allergy. However, ASL has a lower signal-to-noise ratio than DSC and DCE [85]. Moreover, and the scanning time of ASL is longer than that of DCS, and thus is associated with the consequential risk of movement artifacts [86].

In conclusion, PWI provides multiple additional parameters to overcome the intrinsic limitations of conventional MRI. We can determine the blood volume and blood flow and the components of leakage through PWI so as to obtain further information about the tumor vascular system. Perfusion MRI may become a very useful auxiliary means for evaluating glioma after treatment. However, each perfusion technique has its unique advantages and inevitable defects. In clinical work, we need to formulate a reasonable scanning protocol according to the location, size and complexity of the lesion. Considering the long-term follow-up of glioma patients and the toxic and side effects of gadolinium contrast agent, ASL technology may be a good choice. In addition, with the rapid development of nano medicine, a new specific magnetic nano contrast agent was constructed to overcome the shortcomings of gadolinium contrast agent, such as fast clearance rate, leakage, toxic and side effects, and further improve the image resolution of DSC and DCE. We believe that the development of new high-performance contrast agents is also a research hotspot in the future.

### 3.4. Magnetic Resonance Spectroscopy (MRS)

Magnetic resonance spectroscopy (MRS) can complement the anatomical information from conventional MRI to reflect tissue metabolism and biochemical changes at the molecular level. The most crucial and common MRS metabolites include N-acetylaspartate (NAA), a marker of neuronal density and viability and generally decreases in gliomas; total choline (tCho), a marker of cell membrane turnover/cellular proliferation; total creatine (tCr), suggesting altered energy metabolism; lactate (Lac), a product of anaerobic glycolytic metabolism; and lipid (Lip), a marker of necrosis. Typically, TP is characterized by an elevation in Cho and a decrease in NAA. However, RN is characterized by a variable decrease in NAA, lack of significant increase in Cho and the presence of Lip peak. Cho peak and NAA peak may be the most important parameters in distinguishing TP and TRCs.

The utility of single voxel or single slice multi-voxel proton MRS (^1^H-MRS) may lead to incomplete sampling of the neoplasm. Multi-voxel techniques more realistically depict mixed lesions and can help identify surgical targets. Several studies have achieved a good distinction between TP and RN on the basis of Cho/NAA and Cho/Cr ratios. In previous prospective studies, Bulik et al. [87] and Kazda et al. [42] demonstrated that tCho/tNAA is a meaningful parameter, with a sensitivity and specificity of >90% for distinguishing TP and PsP. A meta-analysis reported that MRS is the most promising advanced MRI technique for the treatment response assessment in HGG patients compared with diffusion and perfusion weighted imaging, with a pooled sensitivity and specificity of 91% and 95%, respectively [47]. The latest technology, three-dimensional echo planar spectroscopic imaging (3D-EPSI), can provide metabolite maps with better spatial resolution and lower partial volume effects. In 3D-EPSI, the incorporation of Cho/Cr in the tumor enhancement area and Cho/NAA in the tumor and peritumor area provided a higher accuracy of 93% in distinguishing TP from PsP [88]. Moreover, 3D-EPSI can provide surgeons with more accurate tumor margins, assist in radiation planning, or be used for personalized treatment planning. MRS has a few limitations. Due to partial volume effects, detecting smaller lesions on MRS is challenging. MRS requires a relatively long acquisition time to detect tissue metabolites in the brain tumor. Factors such as scalp and ventricle could contaminate the MRS signal [47]. Additionally, metabolic overlap between RN and TP has been found, identifying the two entities using MRS alone remains difficult. MRS should be combined with other advanced imaging technologies to improve diagnostic accuracy [89].

Through a review of previous studies, we found that the NAA peak, Cho peak, Cr peak and Cho/NAA or Cho/Cr ratio were the most meaningful parameters for identifying TP and TRCs in MRS technology. Additionally, a new 3D-EPSI technology can obtain high-resolution metabolic maps and find other markers that reflect tumor proliferation and tumor metabolism, such as glycine (Gly), glutamic acid (Glu) and glutamine (Gln). Another special metabolite is 2-hydroxyglutarate (2HG), which can provide potential value for MRS to predict IDH status. Therefore, we need to focus on these makers in clinical work and further discover their potential role in evaluating treatment response.

### 3.5. Amide Proton Transfer Imaging

Based on the chemical exchange saturation transfer mechanism, the amide proton transfer imaging (APT) imaging can generate image contrast using endogenous mobile proteins and peptides in tissues, reflecting tumor metabolism [90]. APT asymmetry values are clinically associated with cell proliferation levels. Therefore, APT is widely used as a molecular marker for predicting cellular proliferation and response after tumor treatment. A study reported that among grade II gliomas, IDH wild-type gliomas have higher APT signal strength than IDH mutant gliomas, demonstrating APT as a potential tool for predicting the tumor gene mutations [91].

Many studies have confirmed the ability of APT to distinguish TP and TRCs of gliomas [92,93,94,95]. A few preclinical studies in rat have clearly shown that active glioma and RN exhibit opposite APT signals [92]. Regions with RN generated hypointense or isointense APT signals, while tumors exhibited hyperintense APT signals in the most actively growing tumor areas, which can be readily distinguished. Ma et al. [93] performed a 3D APT sequence scan on patients with clinically suspected TP after chemotherapy and found that the mean APT signal intensity of TP was significantly higher than that of PsP. Another study showed that the APT conversion rate in the lesion area was positively correlated with the Cho/Cr value, both can reflect the cell proliferation state. However, the accuracy of APT in diagnosing TP is 72%, while that of MRS is only 37%. This may be because APT can better reflect the heterogeneity and microenvironment of the entire tumor tissue than MRS, and the reproducibility of ROI selection is low; the selected ROI cannot represent the entire tumor tissue [95]. Both APT imaging and ^14^C-MET-PET reflect endogenous protein metabolism; however, a study found that the diagnostic performance of APT for TP is higher than that of ^11^C-MET PET, as RN and BBB disruption may induce nonspecific MET accumulation [33]. Compared with CE-T1WI and perfusion MRI, Park et al. [94] reported that adding APT imaging to conventional and perfusion MRI improves the diagnostic performance of distinguishing TP and TRCs. Multi-technology integration can improve diagnostic performance. The higher APT signal intensity in TP may be due to the hypercellularity and abundant cytoplasm in tumor cells, whereas the lower APT signal intensity in PsP is probably related to the absence of mobile cytosolic proteins and peptides owing to the cytoplasm loss. In addition to the assessment of post-treated gliomas, APT allows the indirect identification of most active parts of tumors; it can be used to guide stereotactic biopsy [96]. Thus, APT imaging could provide a potential biomarker for brain tumors, increase the accuracy of pathology, and provide more accurate local therapies. However, several challenging technical issues are associated with the use of APT imaging. The APT signals are considered to be affected by many factors, including tissue water content, temperature and tissue acid–base balance [97]. Differences in APT pulse sequences and data processing strategies may complicate the reproducibility and comparison of results between different hospitals [98]. At present, on 3T MR equipment, the APT signal is very weak, which is 2–4% of the water signal. Thus, acquisition and analysis approaches need to be further optimized. This technology has recently become a research hotspot in postoperative treatment evaluation and intraoperative stereotactic biopsy for tumors.

The significance of APTw imaging is that endogenous cellular protein information is obtained indirectly through the bulk water signal used in MRI, thus expanding the range of molecular MRI techniques to the protein level. Currently, in addition to finding suspected recurrent lesions, another major advantage of APT is to guide clinical biopsy and radiotherapy planning. In the long term, this could potentially reduce the necessity for repeated biopsies, and avoiding the associated risks of complications.

### 3.6. Positron Emission Tomography

PET tracer technology can reflect the metabolic information of lesions from different angles and utilize the tissue uptake rate of radioactive materials to identify the TP and TRCs of gliomas. ^18^F-FDG is the most widely available PET tracer. It is based on glycolytic metabolism and has been used to detect the TP and RN in patients with gliomas. Jena et al. studied 35 glioma-treated patients including 41 enhancing lesions with ^18^F-FDG PET/MRI. The accuracy of parameters, such as rCBV_mean_ ADC_mean_, Cho/Cr, maximun tumor-to-brain ratio (TBR_max_), and mean TBR(TBR_mean_) in detecting glioma recurrence were 77.5%, 78%, 90.9% 87.8% and 87.8%, respectively. On multivariate ROC analysis, the maximum AUC was 0.935 ± 0.046 when ADC_mean_, Cho/Cr and TBR_mean_ were combined [31]. However, ^18^F-FDG is limited by its ability to highly accumulate in normal brain tissue, resulting in low contrast between the tumor and normal brain tissue and thus misleading the diagnosis to a certain extent. Numerous promising biomarkers are currently being investigated, such as an amino acid PET tracer O-(2-(18)F-fluoroethyl)-L-tyrosine (^18^F-FET), ^11^C-methyl-L-methionine (^11^C-MET) and^18^F-fluoro-3.4-dihydroxy-L-phenylalanine (^18^F-DOPA).

^18^F-FET, as a promising tracer, is widely used to evaluate the treatment response to gliomas. A study showed that ^18^F-FET PET could identify PsP with 96% accuracy, and ROC analysis showed that the best TBR_max_ cutoff value for identifying PsP was 2.3. In addition, TBR_max_ < 2.3 predicted a significantly longer OS [28]. Other studies have shown that ^18^F-FET could provide valuable information for differentiating TP and TRCs; however, the accuracy of different studies varies widely [30,35,99]. A study used static and dynamic ^18^F-FET uptake parameters to differentiate TP from TRCs of gliomas. Compared with conventional MRI (85%), a higher accuracy (93%) was achieved with ^18^F-FET PET when TBR_mean_ was ≥2.0 and time to peak (TTP) was <45 min was present [29]. In another study, consistent with the previous study, static and dynamic ^18^F-FET PET parameters achieved higher accuracy (93%) than ADC values (69%). With the addition of the static parameter to the ADC value, the diagnostic accuracy of the ADC value increased to 89% [34]. Disadvantages of ^18^F-FET include slower renal elimination, resulting in increased levels of the residual tracer in the blood pool. In a study of 50 patients, the accuracy of hybrid ^11^C-MET PET/MRI in distinguishing TRC and TP in gliomas was significantly higher than that of conventional MRI (96% versus 82%) and ^11^C-MET PET (96% versus 88%) alone, and a TBR_max_ of 1.83 and TBR_mean_ of 1.5 were found to be the optimal cutoff values for distinguishing these two entities [32]. For distinguishing TP in post-treatment HGG patients, APT_max_ and APT90 had a similar to better diagnostic performance than TBR_max_ and TBR90 [33]. In a well-designed prospective study that compared ^11^C-MET with ^18^F-FET, both tracers showed the same sensitivity (91%) and specificity (100%) in differentiating TP from TRCs [100]. Similarly, another promising PET radiopharmaceutical is the ^18^F-FDOPA, which may be highly sensitive and specific for TP detection. Combined analysis of CBF-ASL and ^18^F-DOPA-PET uptake showed the highest specificity (100%) and sensitivity (94.1%) in differentiating TP and PsP in treating gliomas [36]. Amino acid PET is a promising molecular neuroimaging technique that can provide information on tumor metabolism. Table 1 summarizes representative PET studies for detecting TP and TRCs. In the future, the development of glioma specific PET imaging agents and molecular probes will be a research hotspot. In addition, the integration of PET and MRI technology and the integration of tumor molecular metabolism and structural and functional information can further improve the accuracy of distinguishing the TP and TRCs and address crucial clinical problem.

### 3.7. Multi-Model Imaging Modality

Multi-model MRI may be used as a potential surrogate endpoint for TRC and TP assessment. Because each parameter can show different aspects of tumor biology, using a combination of parameters for measurement may have an additional value compared with single-parameter measurement. A meta-analysis [89] showed that for a combination of DWI, PWI and MRS, the pooled sensitivity and specificity in determination of TRCs and TP were 84% (95% CI: 74–91) and 95% (95% CI: 83–99), respectively. With the combination of MR perfusion and diffusion parameters, Nael et al. [101] found that rCBV outperforms ADC and K^trans^ when using a single imaging classifier to predict the two entities, and rCBV and K^trans^ may be used in combination to improve the overall diagnostic accuracy. Some studies have also used the volume-weighted voxel-based multi-parametric clustering (VVMC) method to process MR imaging, and the results showed that VVMC has the highest consistency among observers. Compared with single-parameter measurement, VVMC is a superior and more reproducible imaging biomarker for differentiating PsP and TP in patients with HGGs [102,103]. The addition of MRS to the three perfusion techniques can result in a diagnostic accuracy of 90.0% for evaluating TRCs and TP [73]. Table 2 summarizes the performance of a single technique and a combination of multiple techniques for diagnosing TP and TRCs in recent years. Although studies have shown that multimodal MRI practically improves the diagnostic accuracy of TRCs and TP, most studies, so far, had a samples size and had heterogeneity between lesions. Many studies were retrospective, and the research methods lacked a standardized process. Through the harmonization of collection and post-processing, computer-aided technology and detailed evaluation of clinical and pathological data, diagnostic accuracy may be improved in the future. Table 3 lists the current MRI techniques and their advantages and limitations for differentiating TP and TRCs of gliomas.

## 4. Emerging Application of Artificial Intelligence

Artificial intelligence (AI), machine learning (ML), neural networks (NNs), deep learning (DL) and convolutional NNs (CNNs) are the main components of data science. They have attracted considerable attention in neuro-oncology field by discovering hidden information and reflecting the spatial and temporal heterogeneity of tumors [104]. Exploring multi-omics information, including genomics, transcriptomics, epigenomics, proteomics, metabolomics and metagenomics, has recently become the key to the realization of precision medicine. As an emerging field of medical imaging, radiomics extracts multiple features by using AI methods and has been created to assist in glioma clinical decision management, including to predict tumor grading, potential genetic mutations, survival rates and molecular classification; to automate diagnosis from histopathological slides to segment tissues for surgical planning; and to monitor patients after treatment [105].

### 4.1. Grading and Molecular Information Prediction

With the development of classical ML and DL, the clinical research of AI-based MRI was initiated. Prediction glioma grades using the AI algorithm may have a significant role in future practice, it can evaluate features that are difficult to observe by humans. The distinction between LGG and HGG is crucial for treatment planning and prognosis. In a study of 153 glioma patients, SVM models were established using 30 and 28 optimal features, respectively, to classify LGGs from HGGs and grade III gliomas from grade IV gliomas. The accuracy of classifying LGGfrom HGG was 96.8% and that of classifying grades III from grade IV was 98.1% [106]. By using DL algorithms and transfer learning, a study could predict glioma grades. By including a total of 113 glioma patients, the study found that the AUCs of GoogLeNet and AlexNet were 0.939 and 0.895, respectively. Through transfer learning and fine-tuning, both AlexNet and GoogLeNet have achieved improved performance [107]. In summary, AI algorithms based on imaging features can be feasibly used to predict the glioma grade.

The 2021 WHO CNS classification of brain gliomas clearly defines the new integration diagnostic criteria of multi-molecular classification and precisely describes the malignant biological behavior, which is more conducive for formulating an individual diagnosis and treatment plan [108]. Clinicians attempt to evaluate histopathological features and glioma genomics from non-invasive imaging using AI. Radiomics based on multimodal MRI can precisely differentiate among glioma molecular, including IDH [109], MGMT [110], TERT [111] and H3 K27M [112]. A study established SVM models for detecting IDH and TP53 mutations, and the detecting accuracies for IDH and TP53 mutations on the development cohort were 84.9% and 92.0%, while that on the validation cohort were 80.0% and 85.0%, respectively [113]. Another study showed that based on internal 5-fold cross-validation radiomics of ^18^F-FET PET-MRI performed well in identifying HGGs and LGGs and predicting IDH1, ATRX, 1p19q and MGMT, with AUCs of 88.7%, 85.1%, 97.8% and 75.7%, respectively [114]. Using an NN model, other researchers predicted IDH mutation with an accuracy of 85.7% excluding the patients’ age; when age was incorporated into the model at diagnosis, the accuracy increased to 89.1% [115]. More importantly, radiomics based on the MRI model can help characterize core signaling pathways, thus determining that may determine the type of targeted therapy [116]. AI algorithms are a promising approach analyzing multi-molecular classification and predicting the histopathology of GBM subtypes.

### 4.2. Post-Treatment Follow-Up and Outcome Prediction

Some studies have used AI algorithms to distinguish TP and TRCs. These analysis methods can also be extended to treatment strategies. The SVM classifier has been trained to diagnose PsP and TP in glioma patients undergoing surgery and chemotherapy. In a study of 31 patients, the sensitivity and specificity of the classifier for PsP were 89.91% and 93.72%, respectively, with an AUC of 0.94, and notably, the best predictor image sequences were from DWI and PWI [117]. Another study also found that the SVM classifier is more accurate than experts in identifying TP and RN, with an accuracy of 80% [118]. A multicenter study built a classifier using radiomic features obtained from both K^trans^ and rCBV pharmacokinetic maps coupled with SVM. The study achieved an accuracy of 90.82% in differentiating PsP from TP [119]. The SVM classifier based on the combination of pre-and post-contrast subtraction T2 FLAIR and T1WI imaging also provides a novel idea recognizing TRCs and TP, with an accuracy of 93.33% [120]. Chen et al. attempted to predict PsP using gray-level co-occurrence matrix (GLCM) texture features of conventional MRI and reported an accuracy of 86.4% [121]. Incorporating DWI and PWI, Kim et al. indicated that the multiparametric radiomics model (AUC = 0.90) performed significantly better than any single ADC (AUC = 0.78) or CBV (AUC = 0.80) parameter and than the single radiomics model using conventional MRI (AUC = 0.76) in identifying PsP [122]. CNN, a branch of DL, offers new perspectives for image analysis. Currently, DL algorithms have also been used to identify TP and PsP. A CNNs has been developed to distinguish TP and PsP in patients with GBM status after resection and chemoradiotherapy; it exhibited an acceptable performance and an AUC of 0.83 [123]. Many AI algorithms based on MRI have recently been used to evaluate the therapeutic response of gliomas.

In addition, several studies have focused on using AI algorithms to predict the OS and progression free survival (PFS) of patients with gliomas. By developing aCNN, Lao et al. [124] predicted the survival rate of GBM patients with a C-index of 0.71, which subsequently improved with the inclusion of clinical data. C-index is equal to the AUC and ranges from 0.5 to 1.0, values of >0.7 indicate a good test. Another study demonstrated that a nomogram established using the radiomics signature and clinicopathologic risk factors had high accuracy and good calibration for predicting PFS in both the training (C-index = 0.684) and validation (C-index = 0.823) cohorts [125]. The radiomic signature can stratify patients into low- and high-risk groups. A nomogram combining IDH and age risk factors further improved the performance for predicting OS (C-index = 0.764 and 0.758 in training and test cohorts, respectively) [126]. AI algorithms can supplement the existing diagnostic methods of clinicians through comprehensive and in-depth evaluation of imaging data. In the future, AI can help in the integration of data from different fields (clinical examination, imaging and pathology) to guide treatment and prognosis.

At present, many studies apply AI algorithms to the field of medical image processing, through image preprocessing, image segmentation, feature extraction and model construction, to achieve tumor grading, molecular typing prediction, and efficacy and prognosis evaluation. In addition, AI algorithms also play an important role in surgical navigation, radiotherapy planning and digital pathological grading. ML is a sub-type of AI focused on developing algorithms that can identify patterns within data without explicit specification. Supervised ML algorithms are trained on a human-labeled dataset, which subsequently provide classification or regression on unlabeled data. SVM is one of the most commonly used supervised ML algorithms, and SVM is particularly suitable for handling classification problems. SVM models based on different image features can achieve a variety of classification purposes, and the classification performance is better than other classifiers. The NN is a more complex machine learning algorithm with many variations that attempt to mimic the functionality of biological neural networks and is often used for regression and classification problems. CNN is a typical artificial neural network in deep learning. It has achieved the most advanced performance in image and video recognition and segmentation and can automatically learn deeper and abstract image features in the training process. When the data sets are large enough, the deep learning algorithms often perform better compared to traditional algorithms. However, when it comes to medical image analysis domain, the data sets are often inadequate to reach full potential of deep learning. Transfer learning can solve the problem of insufficient training set caused by the lack of medical image data. It uses natural image data sets or other medical image data sets to pre-train the network, and then transfers the learned knowledge to the target task to improve the network model performance. Therefore, we can choose the appropriate AI algorithm according to different clinical problems and data set size, establish the optimal model and achieve high diagnostic and prediction performance.

### 4.3. Future Challenges

AI has recently provided a new direction for the individual diagnosis and evaluation of tumors. It not only helps radiologists with accurate diagnosis, but also provides useful tools for oncologists in treatment planning and response evaluation. However, certain challenges remain to be addressed in further studies. First, although multimodal images provide comprehensive information on the structure and function of gliomas, the scanning parameters vary across institutes and hospitals, and different scanning protocols may produce different analysis results. Obtaining standardized and large-scale image data to make AI clinically applicable is difficult. Additionally, AI algorithms involve several critical steps. At present, automated intelligent analysis methods that integrate data acquisition, segmentation, feature extraction, modeling and prediction to comprehensively guide glioma diagnosis and treatment are lacking. Especially, in most studies, the lesions were still manually segmented, which is time-consuming, labor-intensive and more subjective. Finally, some clinical experts often regard radiomics as a black box. Thus, the interpretability of radiomics characteristics and AI models needs to be improved urgently. In conclusion, clinically, a large amount of standardized data from multiple centers is required for model testing and verification. Meanwhile, the stability, reproducibility and interpretability of radiomics features and AI models should be considered to make radiomics more acceptable in clinics. More multicenter studies with larger samples are required for external validation in the future.

## 5. Conclusions

Gliomas are the most common and fatal malignant brain tumors in adults. Diagnosis and treatment response evaluation in patients with gliomas are still highly dependent on neuroimaging. Despite active multimodal treatment, the prognosis remains poor. Histopathological examination is the “gold standard” for identifying TP and TRCs; however, it is invasive in nature, which limits its clinical application. The continuous development of neuroimaging technology provides new insights into the potential tumor biology. Dynamic MRI-enhanced scanning can help in effectively identifying TP and TRCs after treatment over time, but it lacks timeliness, and it is difficult to diagnose accurately. Advanced MRI functional imaging techniques can reflect the changes of lesions from different perspectives (cell proliferation, blood perfusion, brain metabolism, etc.). The combination of multimodal MRI techniques can provide multidimensional information in distinguishing two entities (e.g., for patients with good general condition, we try to adopt a variety of MRI techniques to provide complete information of lesions, including diffusion, perfusion, MRS, APT; on the contrary, for patients with poor general condition, we may recommend some techniques with short scanning time, such as DWI and DSC). On this basis, we mine the massive information of multimodal MRI images through radiomics and AI algorithms, which may further improve the accuracy of early diagnosis. Our research group is currently trying to identify TP and TRCs from different perspectives through different MRI techniques and AI algorithms and has achieved certain results. We found that DKI technology can be used as a non-invasive method to identify both, and the MK parameter is a good predictor; at the same time, we also combined multiple MRI techniques to further improve the accuracy of diagnosis. However, so far, there is no consensus between domestic and foreign countries on which MRI technology and AI algorithms to use to identify TP and TRCs. In order to promote widespread clinical acceptance, standardization and harmonization of methodology, guidelines have been provided for data acquisition and analysis, quality evaluation and data interpretation of MR diffusion [127], perfusion [128], spectroscopy [129] and PET131 [130] technology. With the continuous development of 5G technology, AI algorithms also need to be further updated and optimized. Additional improvement in this field requires data sharing and multi-institutional large sample prospective clinical research and validation studies. Finally, a database integrating clinical, pathological, imaging and genetic information is established to achieve precision diagnosis and treatment. With advancement of personalized medicine, these emerging AI-driven neuroimaging technologies may improve quality of life and overall outcomes of these patients.

## Figures and Tables

**Figure 1 cancers-14-03771-f001:**
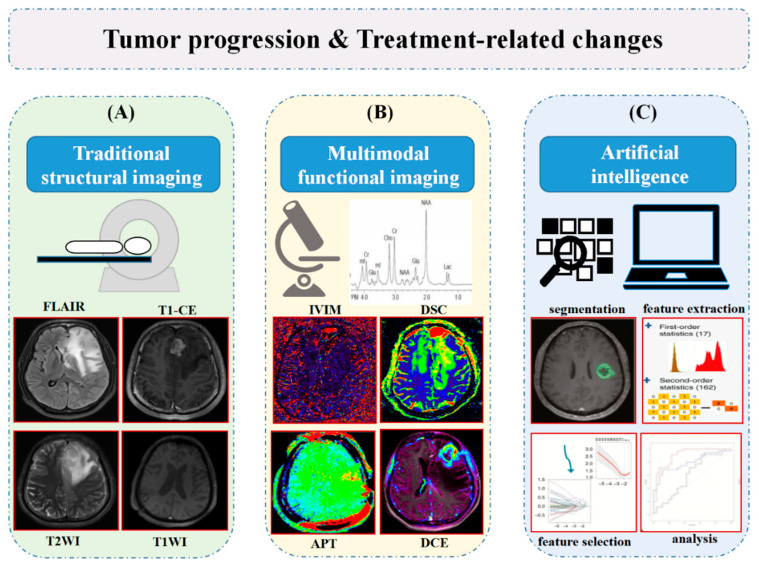
This picture summarizes the basic framework and ideas of the review. We concentrate our discussion mainly on recent advancements from conventional (**A**) and advanced (**B**) imaging to AI (**C**) in identifying TP and TRCs. (**C**) Copyright 2019, OXFORD UNIVERSITY PRESS LICENSE.

**Table 1 cancers-14-03771-t001:** Cut off values of PET tracers for the detection of TP and TRCs.

Study	TP	TRCs	Modality	Tracer	Parameter	Cutoff	Sensitivity	Specificity	Accuracy
Galldiks et al. [28]	11	11	PET	^18^F-FET	TBRmax,	2.3	100%	91%	96%
Galldiks et al. [29]	121	11	PET/MRI	^18^F-FET	TBRmean	2.0	93%	100%	93%
Kebir et al. [30]	19	7	PET	^18^F-FET	TBRmax	1.9	84%	86%	85%
Jena et al. [31]	25	10	PET/MRI	^18^F-FDG	TBRmax TBRmean	1.579 1.179	93.3% 90.0%	72.7% 81.8%	87.8% 87.8%
Deuschl et al. [32]	35	15	PET/MRI	^11^C–MET	TBRmax TBRmean	1.83 1.33	97.14%	93.33%	96%
Park et al. [33]	38	5	PET/MRI	^11^C–MET	TBRmax	1.40	82.1%	66.7%	-
Werner et al. [34]	38	10	PET/MRI	^18^F-FET	TBRmax TBRmean	1.95	100%	79%	83%
Maurer et al. [35]	94	33	PET	^18^F-FET	TBRmax	1.95	70%	71%	70%
Pellerin et al. [36]	34	24	PET/MRI	^18^F-DOPA	Tumor isocontour maps and T-maps	-	100%	94.1%	-

**Table 2 cancers-14-03771-t002:** The diagnostic performance of various technologies for TP and TRCs, and related parameter thresholds.

Study	TP	TRCs	Modality Imaging	Parameter	Cut-off for TP	Sensitivity	Specificity	Accuracy
Lee et al. [41]	10	12	DWI	Mean ADC	1200 × 10^−6^ mm^2^/s	80.0%	83.3%	81.2%
Yoo et al. [25]	24	18	DWI	The 5th percentile of ADC (b = 1000)	915 × 10^−6^ mm^2^/s	83%	67%	-
Chu et al. [43]	15	15	DWI	The 5th percentile of ADC (b = 3000)	645 × 10^−6^ mm^2^/s	93.33%	100%	88.9%
Kim et al. [49]	31	20	IVIM	Mean 90th percentile for perfusion (f) Mean 90th percentile for nCBV	0.056 2.892	87.1% 83.9%	95.0% 95.0%	- -
Kong et al. [68]	33	26	DSC	Mean rCBV	1.47	81.5%	77.8%	-
Baek et al. [70]	42	37	DSC	Skewness and kurtosis of normalized CBV	1.27	85.7%	89.2%	-
Yun et al. [79]	17	16	DCE	Mean K^trans^/mean V_e_	0.347/0.570	59%/88%	94%/56%	-
Yoo et al. [75]	16	8	DCE	Mean V_e_	0.873	100%	63%	88%
Thomas et al. [77]	24	13	DCE	V_p_90%/mean V_p_/mean K^trans^	3.9/3.7/3.6	92%/85%/69%	85%/79%/79%	-
Bisdas et al. [78]	12	6	DCE	K^trans^/IAUC	0.91/15.35	100%/75%	83%/67%	-
Suh et al. [76]	43	36	DCE	mAUCRH/50thAUCR	0.31/0.19	90.1%/87.2%	82.9%/83.1%	-
Chung et al. [72]	32	25	DCE	mAUCRH/90thAUCR	0.23/0.32	93.8%/90.6%	88%/88%	- -
Ma et al. [93]	20	12	APT	APT_mean_/APT_max_	2.42/2.54	85.0%/95%	100%/91.7%	-
Choi et al. [82]	34	28	ASL/DSC	CBF/normalized rCBV	-	94.1%	82.1%	88.7%
Nael et al. [101]	34	12	DWI/DSC/DCE	rCBV/Ktrans	2.2/0.08	94.1	91.6	92.8
Razek et al. [56]	24	18	ASL/DTI	CBF/FA/MD	-	93.8%	95.8%	95%
Seeger et al. [73]	23	17	DSC/DCE/ASL/MRS	normalized rCBV or rCBF /K^trans^/rCBF/Cho/Cr_n_	rCBV ≥ 3.9 or rCBF ≥ 4.1, K^trans^ ≥ 0.08, rCBF ≥ 2.5, Cho/Cr_n_ ≥ 1.89	82.6%	100%	90%
Wang et al. [58]	21	20	DSC/DTI	FA/CL/rCBV_max_	0.55	76%	95%	-
Prager et al. [6]	58	10	DWI/DSC	ADC/normalized rCBV	ADC ≤ 1.49 × 10^−3^ mm^2^/s/rCBV ≥1.27	51.2%	100%	-
Park et al. [102]	45	63	DWI/DSC/DCE	10th percentileof ADC (ADC_10_)/ 90th percentile of normalized rCBV (nCBV_90_)/ 90th percentile of IAUC (IAUC_90_)	ADC_10_ < 1.14 × 10 mm^2^/s/ nCBV_90_ of 3.19/ IAUC_90_ of 19.42/ total cluster score of 5.91	91.1%	90.5%	90.7%

**Table 3 cancers-14-03771-t003:** Summary of current techniques and their advantages and limitations for differentiating TP and TRCs of gliomas.

Imaging Method	Parameters	Pattern Associated with TP	Advantages	Limitations	References
Conventional MRI and TI-CE	No	Corpus callosum involvement; Subependymal enhancement	Widely applied;	Overlapping images	[23,24]
DWI	ADC	Lower mean ADC value	Characterize tissues and pathologic processes at the microscopic level; reflect the high cellularity	Influenced by many factors, such as inflammatory; Ignore the effects of perfusion	[16,48]
IVIM	D D* f	Higher f and D* Lower D	No contrast required; repeatedly acquire images; simultaneous acquisition of diffusion and perfusion parameters	Low cerebral perfusion fraction; susceptibility artifacts; low signal to noise ratio	[49,52,53]
DTI	FA MD	Lower MD and higher FA values	Measured directional variation of water diffusivity	Affected by many factors Susceptibility artifacts b value setting long acquisition time	[27,58]
DSC	rCBV rCBF MTT	Higher rCBV or rCBF value	Widely available; fast acquisition speed and simple post-processing	Poorer spatial resolution; susceptibility artifacts; contrast agent leakage	[73,74]
DCE	K^trans^ V_e_ V_p_ IAUC	Higher K^trans^, V_e_ and V_p_ value	Higher spatial resolution; less susceptible to artifacts	Longer scan time; decreased temporal resolution; complex pharmacokinetic modeling	[72,76,77]
ASL	rCBF	Higher CBF values	No contrast required; less susceptibility artifacts	Low signal-to-noise ratio; risk of movement artifacts	[80,84,86]
MRS	Cho/NAA NAA/Cr Cho/Cr	Higher Cho/NAA and Cho/Cr and lower NAA/Cr	Reflects tissue metabolism; higher diagnostic accuracy	Long scan times required; voxel selection; metabolic overlap	[47]
APT	APTw	Higher APTw signals	Reflect cell proliferation; guide biopsies	Signal weakness; further optimized	[97,98]
^18^F-FDG PET	SUV_TBR_	Higher TBR	Widely available	High background signal	[5]
^11^C-MET PET	SUV_TBR_	SUVs tend to be higher	Lower background activity	Short half-life; requires an on-site cyclotron	[32]
^18^F-FET PET	SUV_TBR_	Higher TBR	High contrast longer half-life efficient synthesis	Requires more research	[28,99]

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
