# Peer review of "Tumor Progression and Treatment-Related Changes: Radiological Diagnosis Challenges for the Evaluation of Post Treated Glioma"

_cancers, 2022, doi:10.3390/cancers14153771_

Round 1

Reviewer 1 Report

This is a review from conventional to functional radiographic imaging approaches for differentiating tumor progression versus treatment-related changes, and provides perspective on the development of artificial intelligence in this endeavor. The authors did a good job covering the mechanism of each technology, as well as corresponding strengths and limitations supported by studies in literature. The manuscript is generally well written and supplemented with informative graph and tables. However, the quality of the paper is limited by lack of information synthesis and high-level comparison of metrics such as “level of evidence” or “clinical readiness” to be of significant value guiding the field in its development (see a review paper on similar topic published recently for example, https://www.frontiersin.org/articles/10.3389/fonc.2021.811425/full). In addition to pointing out weaknesses in each method, the readers would benefit more if the authors could point to potential solutions to mitigate those weaknesses.

Major issue:

For a review article, the studies cited should stay current to the field. Not many references are from the past three years and quite a few a more than 10 years old. The literature cited could be updated to improve relavance of the review.

Since definition of tumor progression evolved over time, it would be useful to discuss with what criteria TP and TRC are defined in the studies reviewed if not all patient had biopsy performed to get gold standard diagnosis.

Specific issue:

Line 52: It’s debated in the neurosurgical field whether standard of care for HGG is “maximal safe resection” or “gross total resection”. It’s better to say “surgical resection” here to avoid controversy.

Author Response

Please see the attachment, thank you!

Reviewer 2 Report

English must be revised.

Lines 54-60: This concept should be reformulated because is based on a 2005 reference and treatment protocols and survival times do not correspond to current guidelines and data. Molecular studies must also be considered in the classification of low grade gliomas.

Line 143: It has not yet been demonstrated that PsP is associated with a better prognosis. Studies are not yet conclusive. This statement should be rephrased.

Line 173: PsP and RN also differ in their pathophysiological mechanism, which should also be highlighted in the text.

Lines 244-248: The definition of progression in the first 12 weeks by the RANO criteria is not correct

Lines 398-415: The data on diffusion kurtosis imaging  are very fragile and do not allow conclusions.

Lines 555-556:  The role assigned to the NAA is not entirely correct, so it should be reviewed.

It is a very extensive review of imaging assessment methods, in many respects in too much detail, but not adding useful information. This aspect must be reviewed.

In the aspect of Artificial Intelligence, several modalities are described, but the interest of its use is not well explained.

In the conclusions, it does not define a guidance algorithm, which is fundamental in clinical practice. In my opinion, this algorithm should be added with a suggested procedure.

Author Response

Please see the attachment, thank you!

Round 2

Reviewer 2 Report

The authors have significantly improved the manuscript. English has been corrected. They gave adequate answers to the questions asked. Improved the review and further discussed the conclusions.